# CT and MRI Key Features of Benign Tumors and Tumor-like Lesions of the Tongue: A Pictorial Review

**DOI:** 10.3390/cancers17101695

**Published:** 2025-05-18

**Authors:** Michele Pietragalla, Emanuele Gattuso, Cosimo Nardi, Antonio Lo Casto

**Affiliations:** 1Department of Radiology, San Jacopo Hospital, Via Ciliegiole 97, 51100 Pistoia, Italy; 2Department of Biomedicine, Neuroscience and Advanced Diagnostics (BIND), Section of Radiology, University Hospital of Palermo, 90127 Palermo, Italy; emanuelegattuso96@gmail.com (E.G.); antonio.locasto@unipa.it (A.L.C.); 3Department of Experimental and Clinical Biomedical Sciences, University of Florence, Careggi Hospital, Largo Brambilla 3, 50134 Florence, Italy; cosimo.nardi@unifi.it

**Keywords:** benign tumors, computed tomography, magnetic resonance imaging, tongue, tumor-like lesions

## Abstract

Benign tumors and tumor-like lesions of the tongue, though relatively rare, present distinctive radiological characteristics that are crucial for accurate diagnosis and differentiation from malignant tumors. Benign lesions typically maintain a clear delineation from adjacent muscle and bone, with no evidence of deep tissue invasion or regional lymph node involvement—features commonly seen in malignancy. Benign tongue lesions typically appear as well-circumscribed, homogeneous masses with smooth, well-defined borders. On computed tomography imaging, these lesions exhibit a consistent attenuation profile, often like that of surrounding muscle or fat, and are usually devoid of internal necrosis or hemorrhage. Venous malformations and dermoid cyst may show calcifications/phleboliths. On magnetic resonance imaging, benign tumors generally exhibit low to intermediate signal intensity (SI) on T1 and high SI on T2 images, consistent with their soft tissue and/or cystic composition. Lipoma and angiomyolipoma contain fat tissue and show high T1 and T2 SI, and low SI on fat-saturated sequences. Vascular malformations usually show high T2 SI. Lesions such as fibromas, papillomas (both entities not shown in the current review), “dermoid cyst”, and lipomas tend to show minimal contrast enhancement, reflecting their relatively avascular nature compared to malignant tumors. However, schwannomas usually show vivid contrast enhancement. Vascular malformations may demonstrate variable enhancement patterns depending on their composition. Arteriovenous malformations usually show a prominent contrast enhancement due to their rich vascular supply and flow voids due to intralesional high-flow vascular structures.

## 1. Introduction

Benign tumors and tumor-like lesions of the tongue are relatively uncommon and encompass a broad spectrum of pathologies that vary in etiology, anatomical location, and radiological appearance (Table 1) [1]. 

These lesions are generally non-neoplastic or exhibit no potential for malignant transformation [2]. A notable exception is the pleomorphic adenoma, which carries a recognized—albeit rare—risk of malignant transformation into carcinoma ex pleomorphic adenoma [3]. Nevertheless, the occurrence of pleomorphic adenomas arising specifically within the tongue is exceedingly rare [1].

Clinically, most benign tongue tumors present as submucosal protrusions situated within the deep muscular compartments of the tongue. While frequently asymptomatic, they may occasionally cause pain, bleeding, or functional impairment, particularly involving speech and swallowing [2]. Computed tomography (CT) and magnetic resonance imaging (MRI) are mandatory for the diagnostic approach of benign tumors or tumor-like lesions of the tongue, delineating tumor location and extension [4]. CT and MRI may also provide morpho-structural features of these lesions that can be used to narrow the differential diagnosis, suggesting specific histologic tumor types [5]. 

Despite the clinical relevance of these lesions, the existing literature dedicated to their imaging characteristics remains limited, potentially hindering diagnostic familiarity. The benign tumors and tumor-like conditions addressed in this review include schwannoma, lipoma, angiomyolipoma, vascular malformations, dermoid cysts, and remnants of the thyroglossal duct, including both cystic formations and ectopic thyroid tissue. Additionally, non-neoplastic conditions such as lingual abscesses and infectious mononucleosis with secondary lingual tonsillitis can present as mass-like lesions of the tongue and are therefore included in this pictorial review. 

This pictorial review aims to enhance diagnostic awareness by illustrating the characteristic CT and MRI findings of benign lingual lesions (Table 2). A thorough understanding of their imaging profiles may enable radiologists and clinicians to more accurately identify these lesions, thereby minimizing diagnostic uncertainty. Early and precise characterization not only supports optimal therapeutic decision-making but also contributes to improved patient outcomes. 

## 2. Benign vs. Malignant Tongue Lesions: Diagnostic Imaging Features

CT and MRI play a critical role in distinguishing between benign and malignant lesions of the tongue, providing essential information for diagnosis, staging, and treatment planning. Benign lesions such as fibromas, papillomas (both entities not shown in the current review), lipomas, angiomyolipoma, and venous malformations typically appear as well-circumscribed, smooth masses with homogeneous density on CT or MRI scans. These lesions are often located in the anterior or lateral aspects of the tongue and lack signs of invasion into adjacent structures. Venous malformations and dermoid cysts may show calcifications/phleboliths.

On MRI, benign tumors generally exhibit low to intermediate SI on T1 and high SI on T2. Lipoma and angiomyolipoma contain fat tissue and therefore show T1 and T2 high SI, and low SI on fat-saturated sequences. Vascular malformations may show prominent contrast enhancement after administration of contrast media agents [20].

Conversely, malignant tumors, particularly squamous cell carcinoma, exhibit more aggressive imaging characteristics. On CT, they may present as a heterogeneous mass with ill-defined borders, often associated with invasion into nearby soft tissues or bone [21]. MRI provides higher sensitivity for detecting soft tissue invasion and may reveal irregular, ulcerated masses with associated lymph node enlargement. Malignant neoplasms generally show solid tissue with intermediate T2 SI, whereas intratumoral areas of necrosis and cystic changes may manifest as high T2 SI. The invasion of the floor of the mouth or the mandibular bone is frequently observed in advanced cases. Additionally, the presence of regional lymphadenopathy on imaging can be indicative of metastasis, a common feature in malignant tongue tumors [22].

Biopsy remains essential for definitive diagnosis, but imaging findings provide a strong basis for clinical suspicion and help guide further management strategies.

## 3. Schwannoma

Schwannomas are benign, encapsulated, and slow-growing neoplasms that originate from Schwann cells—the myelin-producing glial elements of the peripheral nervous system. They most commonly present between the second and fourth decades of life, without a predilection for gender or ethnicity. Although schwannomas may develop at any anatomical site, approximately 25–40% arise in the head and neck region, with the carotid space representing the most frequently affected location. Lingual schwannomas are uncommon, typically involving the hypoglossal, glossopharyngeal, or lingual nerves [23]. Clinically, lingual schwannomas often present as well-circumscribed, elastic, and painless masses covered by intact mucosa. However, ulceration and tenderness may occasionally be observed [24]. 

Schwannoma appears as a solid, round well-defined mass, hypodense or isodense to the muscle on CT. Given their high cellularity and absence of intralesional fat, magnetic resonance imaging (MRI) offers superior diagnostic characterization (Figure 1). On MRI, schwannomas demonstrate intermediate to low signal intensity (SI) on T1 and high SI on T2 and fat-suppressed sequences. Following intravenous contrast administration, they usually exhibit intense and homogeneous enhancement. In cases of intralesional cystic or fatty degeneration, enhancement may be heterogeneous [6]. Several characteristic MRI signs can assist in the radiologic diagnosis of schwannoma. These include the “split fat sign” (a thin peripheral rim of fat surrounding the lesion), the “target sign” (central low SI within the lesion surrounded by higher peripheral SI), and the “fascicular sign” (multiple small internal ring-like structures suggestive of nerve fascicles) [7].

*Differential diagnosis:* Slow and homogeneous filling following intravenous injection of contrast medium and intralesional phleboliths should raise the suspicion of venous malformation [9]. Dermoid and epidermoid cysts, in contrast, do not enhance following contrast administration. Dermoid cysts contain sebaceous and lipid-rich material and may demonstrate free calcified corpuscles and/or clusters of small fat globules—the so-called “sack of marbles sign”. On CT, the fatty components appear hypodense, similar to subcutaneous adipose tissue, whereas calcified corpuscles are hyperdense. On MRI, intralesional fat demonstrates high SI on both T1 and T2 sequences, while calcified corpuscles are hypointense on T2 images [13]. Epidermoid cysts characteristically exhibit high SI on diffusion-weighted imaging (DWI) and low signal on apparent diffusion coefficient (ADC) maps, consistent with restricted diffusion [25]. In contrast, lipomas are composed exclusively of mature adipose tissue and exhibit homogeneous high SI on T1 MRI, which is suppressed on fat-saturated sequences, without post-contrast enhancement.

## 4. Lipoma and Angiomyolipoma

Lipoma and angiomyolipoma are benign, slow-growing lesions characterized histologically by the presence of mature adipocytes. In the case of angiomyolipoma, these adipocytes are admixed with thick-walled blood vessels and interspersed bundles of smooth muscle fibers [26]. Lipoma is the most common benign neoplasm of adipose tissue, accounting for approximately 15–20% of benign tumors in the head and neck region; however, its prevalence in the oral cavity is considerably lower, estimated at 1–4% [27]. Clinically, oral lipomas present as soft, often asymptomatic masses, typically covered by normal or yellowish mucosa. They are usually less than 2 cm in diameter and may arise from a variety of intraoral sites, including the buccal mucosa, tongue, lips, retromolar area, floor of the mouth, and gingiva [28,29]. In the tongue, the most common location is within its anterior two-thirds. 

CT and MRI features of lipomas are dictated by their composition of mature adipose tissue. Simple lipomas—representing approximately 80% of all histologic subtypes—appear as homogeneous, well-circumscribed masses with fat attenuation on CT (ranging from −83 to −134 Hounsfield units) and with signal intensity comparable to subcutaneous fat on MRI (i.e., high SI on T1 and T2 sequences with homogeneous suppression on fat-saturated sequences). These lesions do not enhance following intravenous contrast administration [8] (Figure 2).

Spindle cell lipoma is a rare variant composed of a mixture of mature adipose tissue and spindle-shaped non-adipose stromal cells. This histologic complexity results in more heterogeneous imaging findings compared to simple lipomas. The non-fatty stromal components of spindle cell lipomas typically exhibit intermediate SI on T1 images and mildly increased SI on T2 images, while the fatty components demonstrate uniformly high SI on both sequences [30,31]. Post-contrast enhancement is typically mild and heterogeneous, and significantly less than that of adjacent normal tongue tissue [1]. Oral angiomyolipomas are exceedingly rare, with only five cases reported in the literature to date. These lesions have been identified in individuals between the third and seventh decades of life, without a clear gender predilection [26]. Similar to spindle cell lipomas, angiomyolipomas demonstrate heterogeneous imaging characteristics due to the presence of mixed vascular, muscular, and adipose tissue components. Contrast-enhanced imaging typically reveals variable enhancement patterns depending on the relative proportions of each component (Figure 3). 

*Differential diagnosis:* Dermoid cysts, although often located in the midline, may occasionally be considered in the differential diagnosis due to their fat-containing components. These lesions exhibit high SI on both T1 and T2 sequences, attributable to the presence of lipid-rich material [1]. Chronic hypoglossal denervation is characterized by progressive reduction in volume and increase in fatty infiltration of the half tongue in the affected side, without a mass-like lesion resulting hypoattenuation on CT, high SI on T1 and T2, and low SI on fat signal saturation sequences [19].

## 5. Vascular Malformations and Tumors 

Vascular anomalies are a heterogeneous group of congenital blood vessel disorders more typically referred to as “birthmark”. According to the Mulliken and Glowacki classification system, first proposed in 1982 and later adopted in the 2018 International Society for the Study of Vascular Anomalies (ISSVA) classification, vascular anomalies are broadly divided into two major categories: vascular tumors and vascular malformations [32]. This distinction is primarily determined by their natural history and behavior, cellular turnover, histological features, and management approach [33]. 

Hemangioma is the most prevalent type of soft tissue tumor in the head and neck region during infancy and childhood, accounting for approximately 7% of all benign soft tissue tumors [34]. Hemangioma originates from the endothelial cells and shows a distinctive growth cycle: a proliferation phase of early rapid growth followed by an involutional phase of slow regression [35], accompanied by fibrofatty infiltration and low mast cell counts by adolescence [36]. 

Vascular malformations are non-neoplastic lesions characterized by abnormalities in the formation of blood and lymphatic vessels. They may be classified as a simple or combined vascular malformation, according to the predominant type of anomalous vessel involved. Simple vascular malformations comprise low-flow lesions (e.g., capillary, venous—also broadly termed venous hemangioma—and lymphatic malformations) and high-flow lesions (e.g., arteriovenous malformation (AVM), arteriovenous fistula) [37]. High-flow malformations always have an arterial component along with variable combinations of slow-flow elements [38]. Combined vascular malformations sharing features of multiple types of vessels may also occur, of which the most frequent is venolymphatic malformation. Unlike hemangiomas, vascular malformations are typified by normal endothelial cells and normal numbers of mast cells throughout their natural history. 

Since “flow voids” are generally absent in low-flow vascular lesions, MRI is useful in the distinction between high-flow and low-flow vascular abnormalities. A typical “flow void” in high-flow lesions is observed as loss of signal in vessels containing vigorously flowing blood in sequences with a long time of echo (T2 and T2*, in particular). In contrast, low-flow lesions are characterized by a low SI in T1 sequence, accompanied by moderate and homogeneous increased SI in T2 images [1].

Although congenital, vascular malformations may not be clinically evident until late infancy or childhood. The growth of these lesions is proportional to that of the patient, and they do not regress or involute. The potential etiologies of accelerated growth in such cases may include trauma, infection, or endocrine changes.

Vascular malformations of the tongue may result in hemorrhage, asymmetric macroglossia, airway compromise, dysphagia, masticatory and phonatory difficulties, and dentofacial anomalies [39,40]. 

### 5.1. Venous Malformations

Lingual venous malformations are typically characterized by tortuous, compressible, and often asymptomatic veins. Clinically, they often appear as solitary, reddish to purplish lesions and may involve the dorsal and ventral aspects of the mobile tongue as well as the tongue base, with no specific predilection for anatomical site [39,41]. However, the tongue tip is frequently involved, and in some cases, the entire tongue or extra-lingual regions may be affected. 

On MRI, venous malformations of the tongue manifest as solid masses with iso or slightly high SI relative to muscle on T1 images, and heterogeneous SI on T2 sequences. Internal features such as phleboliths, flow voids, or septations may appear as discrete hypointense foci or linear strands [9]. Phleboliths are also detectable on CT imaging [1]. Contrast-enhanced MRI typically demonstrates slow, homogeneous enhancement following intravenous administration of gadolinium [9] (Figure 4 and Figure 5).

*Differential diagnosis:* Schwannomas may present with intralesional cystic and fatty degeneration, resulting in an inhomogeneous appearance. Diagnostic signs may include the “split fat sign” (thin rim of surrounding fat), “target sign” (central low SI), and “fascicular sign” (multiple internal ring-like structures) [6,7]. Dermoid and epidermoid cysts exhibit minimal or peripheral enhancement, while lipomas demonstrate fat signal characteristics throughout [14].

### 5.2. Lymphatic Malformations

Lymphatic malformations encompass a wide spectrum of rare congenital abnormalities (1:4000 live births [10]) arising from developmental errors in lymphatic embryogenesis [11]. These lesions predominantly occur in the head and neck and usually present at birth or during early childhood as solitary masses that grow proportionally with the body. Adult-onset presentations, although rare, have been reported [10]. Lymphatic malformation mass lesions are defined as low-flow vascular malformations consisting of dilated lymphatic vessels without cellular atypia, manifesting as a multilocular or unilocular cyst-like lesion with fibrous septa. Three subtypes of lymphatic malformation have been described based on morphologic features, namely, the macrocystic (previously known as cystic hygroma), microcystic (previously known as lymphangioma, Figure 6), and combined subtypes [10,11]. The macrocystic subtype is characterized by locules larger than 1 cm, the microcystic subtype comprises smaller (typically sub-centimeter) cysts, and the combined subtype shares features of both. 

MRI findings include multiloculated cystic lesions with fluid signal characteristics and no flow voids. Hemorrhage or infection may alter signal characteristics and lead to fluid–fluid levels. Mild septal or capsular enhancement may be seen, which becomes more pronounced in the setting of inflammation. Importantly, no solid enhancing components should be present. Fat-suppressed T2 imaging is helpful in detecting perilesional inflammatory changes, which appear only during active inflammation [42]. 

Differential diagnosis: combined venolymphatic malformations share characteristics of both lymphatic and venous malformations (fluid–fluid levels and phleboliths). Dermoid cyst shows free calcified corpuscles, appearing hyperdense on CT and exhibiting low SI on T2 images, and/or intracystic aggregations of multiple small globules with fat attenuation on CT, and high T1 and T2 SI on MRI (“sack of marbles sign”) [13]; epidermoid cyst may show high SI on DWI and low ADC values [25].

### 5.3. Arteriovenous Malformations

Extracranial arteriovenous malformations (AVMs) of the tongue are rare and may present differently from low-flow vascular malformations. Clinical signs include soft tissue swelling with palpable thrill and audible bruit, as well as pain, ulceration, dysphagia, and dysphonia [43]. Lingual AVMs carry a high risk of severe, spontaneous hemorrhage with potentially life-threatening consequences if untreated. MRI provides a good depiction of the typical signal flow voids in both T1 and T2 sequences, resulting in the appearance of serpentine images (Figure 7a–c). MR angiography, CT angiogram, or digital subtraction angiography should be performed to confirm the diagnosis (Figure 7d–h) [12]. The presence of arterial feeding vessels and a “nidus” is a hallmark of AVMs [6]. The term “nidus” means a net-like tangle of multiple small arteries and veins that presents as a localized proliferation of multiple, smaller vessels (“bag of worms”).

Management should be considered even in asymptomatic patients. Therapeutic options include surgical excision, endovascular embolization, radiation therapy, and laser ablation [43].

*Differential diagnosis:* The presence of both phleboliths and flow void in a vascular lesion is indicative of venous malformation combined with AVM [44].

## 6. Dermoid Cysts

Dermoid cysts are benign congenital or developmental lesions that typically arise along the midline due to the sequestration of ectodermal elements during embryologic fusion of the branchial arches. Although uncommon in the head and neck region, they account for approximately 7% of all dermoid cysts, with predilection sites including the lateral third of the eyebrow and the floor of the mouth, and less frequently, the tongue [45,46]. Within the oral cavity, dermoid cysts are exceedingly rare, representing less than 0.01% of all oral cystic lesions, and are usually diagnosed at birth or during early infancy [47]. They affect both sexes equally. However, when located at the ventral base of the tongue, dermoid cysts often present later, during childhood or early adulthood [45]. In contrast, acquired dermoid cysts develop secondary to traumatic implantation of epithelial elements and are typically found off the midline [48]. Both dermoid and epidermoid cysts are ectoderm-derived inclusion cysts; they differ in histological complexity, with epidermoid cysts lined solely by squamous epithelium, whereas dermoid cysts also contain dermal appendages such as sebaceous glands, sweat glands, and hair follicles of endodermal origin [49]. Clinically, dermoid cysts are often asymptomatic and discovered incidentally as painless, slowly enlarging, unilocular cystic masses beneath normal-appearing mucosa. However, in some cases, significant growth can lead to functional impairment, including difficulties with speech articulation, mastication, deglutition, and, in extreme cases, airway obstruction [50]. Dermoid cysts imaging features are variable due to the complexity of their content. The sebaceous or lipid material within a dermoid cyst exhibits SI characteristics that mimic those of fat on CT and MRI, manifesting with high SI on T1 and T2 images [1]. Additionally, dermoid cysts may contain free calcified corpuscles which appears hyperdense on CT and exhibits low SI on T2 images, and/or aggregations of multiple small globules of fat. This latter feature is referred to as the “sack of marbles sign”: multiple intracystic “spheres” with fat attenuation on CT, and high T1 and T2 SI on MRI [13] (Figure 8). In contrast to dermoid cysts, epidermoid cysts lack internal calcifications and macroscopic fat; they may also exhibit significant intralesional diffusion restriction of water molecular motion on DWI sequence with very low ADC values (Figure 9). However, absence of intralesional restricted diffusion in epidermoid cysts is also reported in the literature. Both dermoid and epidermoid cysts show no enhancement or faint linear peripheral enhancement [14]. 

Cystic teratomas are true neoplasms characterized by the presence of tissue structures of mesenchymal origin. They are usually multiloculated and more heterogeneous lesions, typically reflecting more than one of the three embryonic germ layers, namely, ectoderm, endoderm, and mesoderm layers [49].

The radiological appearances of dermoid cysts, epidermoid cysts, and teratoid cysts can overlap significantly. Therefore, histological examination is crucial in making an accurate diagnosis and distinguishing between these three types of alterations [50].

*Differential diagnosis:* Venous malformation contains phleboliths [1] and enhances [9], lymphatic malformation may show fluid-fluid levels [42], and lipoma is made of fat.

## 7. Thyroglossal Duct Remnants

Thyroglossal duct remnants are congenital midline anomalies resulting from aberrant embryologic development and migration of the thyroid gland. These remnants may persist anywhere along the tract of the thyroglossal duct, extending from the foramen cecum at the base of the tongue to the thyroid bed in the lower neck [51]. They may manifest as cystic lesions or as ectopic thyroid tissue.

### 7.1. Thyroglossal Duct Cyst

Thyroglossal duct cysts represent the most common congenital anomaly of the neck. They are typically present at birth, though at least half of them is diagnosed later in life in the second decade or in adulthood. The average patient age at diagnosis is 36 years. About 61% of thyroglossal duct cysts occur at the thyrohyoid level, while only 2% may involve the foramen cecum at the tongue base [52]. Thyroglossal duct cyst usually presents as a mobile, soft, and painless mass in the midline of the neck, in proximity to the hyoid bone. 

CT findings of a thyroglossal duct cyst are represented by a well-circumscribed, low-density lesion with a thin and smooth rim. An enhancement and thickening of the cyst wall and septations and an increase in the density of the cyst content may suggest an additional inflammation or infection of the cyst. On MR images, a thyroglossal duct cyst usually appears as a simple cyst with high T2 SI and low to high T1 SI depending on the degree of proteinaceous or hemorrhagic content [15] (Figure 10).

*Differential diagnosis:* Lingual tonsil mucous retention cysts are usually localized off-midline and more laterally than thyroglossal duct cysts (Figure 11).

### 7.2. Ectopic Thyroid Tissue

Ectopic thyroid tissue is a far less common finding than cystic remnants, accounting for approximately 5% of thyroglossal duct anomalies. Ectopic thyroid tissue may be identified as discrete nodular foci (Figure 12) or embedded within the wall of a thyroglossal duct cyst. While frequently asymptomatic, ectopic thyroid tissue remains susceptible to the same pathologies as orthotopic thyroid tissue, including malignancies. The reported prevalence of differentiated thyroid carcinoma arising within a lingual thyroid or thyroglossal duct cyst is approximately 1% [16]. Imaging features suggestive of malignancy include rapid lesion growth in the absence of infection, irregular or heterogeneous solid components, abnormal vascular patterns, or calcifications [52]. Multiple ectopic thyroids may be observed in the same patient (Figure 13). The lower neck should always be checked for an orthotopic thyroid. TC-99m pertechnetate or I-131 or I-123 scintigraphy is the most important method to determine the presence of functional ectopic thyroid tissue [53,54], and it is useful to confirm the diagnosis of ectopic thyroid. 

*Differential diagnosis:* Squamous cell carcinoma and lymphoma of the tongue base may resemble lingual ectopic thyroid tissue on imaging. However, the absence of adenopathy and similar post-contrast enhancement of the lingual nodule in the same way as the thyroid gland should raise suspicions of lingual ectopic thyroid tissue.

## 8. Lingual Abscess

Lingual abscess is an exceptionally rare disease seldom encountered in the modern era [55]. Elderly and/or debilitated patients are at greater risk of developing a tongue abscess. It is a potentially life-threatening complication of bacterial infection that can be challenging to diagnose in the absence of clinical signs. Predisposing factors include breaches in the mucosal surface, foreign bodies, trauma, and immunodeficiency.

Most lingual abscesses present as a unilateral, submucosal mass located in the anterior two-thirds of the tongue, typically within the muscular tissue [56]. Abscesses originating from the posterior third of the tongue, however, are often associated with lingual tonsillar involvement, infected thyroglossal duct cysts, or as extensions of apical infections from the first or second molar teeth. Clinically, acute tongue abscesses present with rapid onset swelling or a palpable mass in the deep tissues of the tongue, accompanied by pain radiating towards the ears, throbbing local pain, fever, dysphagia, and a voluntary fixation of the tongue due to pain. Respiratory distress and alterations in the blood cell count may also be present [57]. Lingual abscesses may manifest as solid or cystic lesions with sharp margins, perilesional edema, and rim enhancement both on CT and MRI.

On MRI (Figure 14), the core of the abscess has typically low SI on T1, high SI on T2, and high SI on DWI images with very low values on ADC maps, revealing intralesional restricted diffusion of water molecules motion. The peripheral wall of the abscess shows high SI on T1 and low SI on T2 sequences, and rim enhancement after contrast media injection. The presence of a hypointense halo surrounding the wall (“target sign”) on post-contrast T1 images, along with a perilesional hyperintense area of edema on T2 sequence that diffusely enhanced after contrast media administration, may suggest a concomitant perilesional cellulitis [17]. It is important to differentiate between abscess and cellulitis, as the management of the former typically requires drainage, whereas the latter can be effectively managed with antibiotics alone [58].

*Differential diagnosis:* In rare cases, lingual abscess may occur with slowly progressing and subtle symptoms, nuanced objective and laboratory findings, and inconclusive radiological evidence, leading to difficult differential diagnosis with submucosal malignancy [56]. In these cases, biopsy is needed to confirm the diagnosis [59]. Lingual abscess may be infrequently induced by tongue squamous cell carcinoma [57].

## 9. Lingual Tonsillitis as Manifestation of Infectious Mononucleosis

Infectious mononucleosis primarily affects adolescents and young adults between 15 and 24 years old [60], but it can also affect elderly people [61]. Symptoms include fever, general discomfort, pharyngitis, and lymphadenopathy. It is caused by Epstein–Barr virus (EBV), which has an incubation period of 4–8 weeks. Diagnosis is made based on clinical manifestations and laboratory findings [62]. 

The lingual tonsils are located at the base of the tongue and extend downward from the circumvallate papillae to the root of the epiglottis. In case of lingual tonsillitis in association with infectious mononucleosis, the lymphoid tissue of the tongue base may become enlarged or hypertrophied, showing distal lingual compression of the epiglottis, effacement of the vallecula, and impingement on the aryepiglottic folds (Figure 15). Due to their distal pharyngeal location, acute inflammation of lingual tonsils can result in moderate to severe dysphagia, odynophagia, and speech alteration [63].

Symptoms associated with lingual tonsillitis are non-specific and may be indicative of numerous other diseases that may cause lingual tonsil enlargement, such as tumors, trauma, abscesses, hyperkeratosis, cysts from occlusion of the foramen cecum, foreign bodies, or compensatory hyperplasia after palatine tonsillectomy. CT and MRI imaging may reveal an enlarged lingual tonsil that demonstrates contrast enhancement, along with mild stranding of the pre-epiglottic fat and bilateral cervical lymphadenopathies [62]. Symmetric enlargement, retention cysts, and “linear or columnar pattern” of enhancement represent the lingual tonsil septa interposed with thickened mucosa and are typical features of benign tonsillar hypertrophy [18]. 

*Differential diagnosis:* An asymmetric lingual tonsil enlargement in absence of a linear pattern of enhancement and/or in the presence of bulky adenopathy may suggest the presence of non-Hodgkin lymphoma of the tongue base [64].

## 10. Fatty Atrophy of the Tongue (Chronic Hemilingual Denervation)

Fatty atrophy of the tongue represents a rare form of chronic neurogenic myopathy, resulting from prolonged denervation of the lingual musculature—typically following injury to the hypoglossal nerve. This condition is most frequently unilateral and often reflects an acquired etiology, particularly in the context of surgical, traumatic, or procedural insult to cranial nerve pathways. Clinicians should maintain a high index of suspicion for this entity in postoperative patients presenting with tongue deviation, dysarthria, or unilateral tongue atrophy, especially in the absence of a mass-like lesion. When associated with concurrent vagus nerve involvement, it is classified as Tapia’s syndrome, a rare complication marked by combined paralysis of the tongue and laryngeal muscles [65].

Among the principal causes, mechanical or positional cranial nerve injury during orotracheal intubation and cervical spine or carotid surgery stands out. Excessive hyperextension or lateral flexion of the neck during prolonged general anesthesia has been implicated as a transient but damaging factor for nerve stretching or compression. Additionally, direct trauma during surgical approaches to the skull base, parapharyngeal space, or cervical neurovascular structures may result in focal neuropathy. Less frequently, skull base lesions, neoplasms, or inflammatory processes such as an inflammatory pseudotumor may be responsible, although these typically present with a more complex constellation of symptoms and imaging findings.

MRI is the preferred modality to evaluate fatty degeneration of the tongue. Classic imaging features include very high SI of the involved hemilingual muscles on T1 sequence, reflecting fat infiltration (Figure 16a). T2 images may also show high SI but are generally less specific. CT depicts chronic hemilingual atrophy with the same density as the subcutaneous fat tissue [19]. The muscles involved exhibit volume loss, with thinning best appreciated on coronal and sagittal views (Figure 16b,c). The process respects the midline, with preservation of contralateral musculature, distinguishing it from diffuse myopathies or systemic infiltrative conditions. Importantly, no enhancement is observed following gadolinium administration, further supporting a chronic, non-inflammatory process. 

*Differential diagnosis:* Lipoma and angiomyolipoma are focal lesions composed of mature adipose tissue, and they do not cause a reduction in lingual volume or a deviation of the tongue.

## 11. CT X-Ray and MRI Scans Impact on Human Head/Thorax Portion

CT scans, particularly those involving the head and thorax, expose the human body to substantial doses of ionizing radiation. X-rays used in CT imaging can penetrate tissues and provide detailed anatomical views, but they also deposit energy into cells, which may damage DNA and increase the long-term risk of cancer. The head and chest regions are especially sensitive due to the proximity of critical organs such as the brain, eyes, lungs, and thyroid. A single high-resolution head or thorax CT scan can deliver a radiation dose equivalent to hundreds of chest X-rays, raising concern over cumulative exposure in patients undergoing repeated scans [66]. A recent study reveals that CT scans may account for up to 5% of all new cancer cases annually in the United States, estimating over 100,000 additional cancer cases linked to CT scan radiation exposure in 2023 alone [67]. 

In contrast, MRI does not involve ionizing radiation. MRI uses powerful magnetic fields and radio waves to generate images of soft tissues, making it a safer option for repeated use. However, the strong magnetic fields and changing gradients can cause temporary side effects such as dizziness, nausea, or metallic taste. In MRI, SAR refers to the amount of radiofrequency energy absorbed by the body’s tissues. High SAR levels, especially during prolonged or high-field MRI scans, can lead to heating of tissues. While MRI is generally considered safe, excessive thermal effects could cause discomfort or tissue damage, particularly in sensitive areas such as the skin, eyes, or peripheral nerves. However, the risk is minimized by strict guidelines that limit SAR levels to prevent significant temperature rise. In clinical practice, MRI scanners are designed to ensure that SAR remains within safe thresholds to avoid harmful effects [68]. As awareness grows regarding radiation risks, alternative diagnostic techniques are being explored. One such approach involves the detection of ultraweak biophoton emissions from living tissues. Pioneered by Fritz Albert Popp and others, this method investigates the spontaneous emission of light from biological systems, potentially revealing pathological changes, including early signs of cancer. Though still largely experimental, biophoton-based diagnostics offer a non-invasive, radiation-free method of assessing tissue function and health, pointing toward a future of safer medical imaging [69,70].

## 12. Conclusions

Benign tumors or tumor-like lesions of the tongue are relatively uncommon and encompass a diverse array of pathological entities. CT and MRI are indispensable imaging modalities in the diagnostic evaluation of these lesions. By interpreting the imaging characteristics, radiologists can narrow the differential diagnosis and provide guidance for the appropriate histological investigation. A comprehensive understanding of typical imaging signs on CT and MRI can help avoid unnecessary invasive procedures, thus enhancing diagnostic accuracy and patient care.

## Figures and Tables

**Figure 1 cancers-17-01695-f001:**
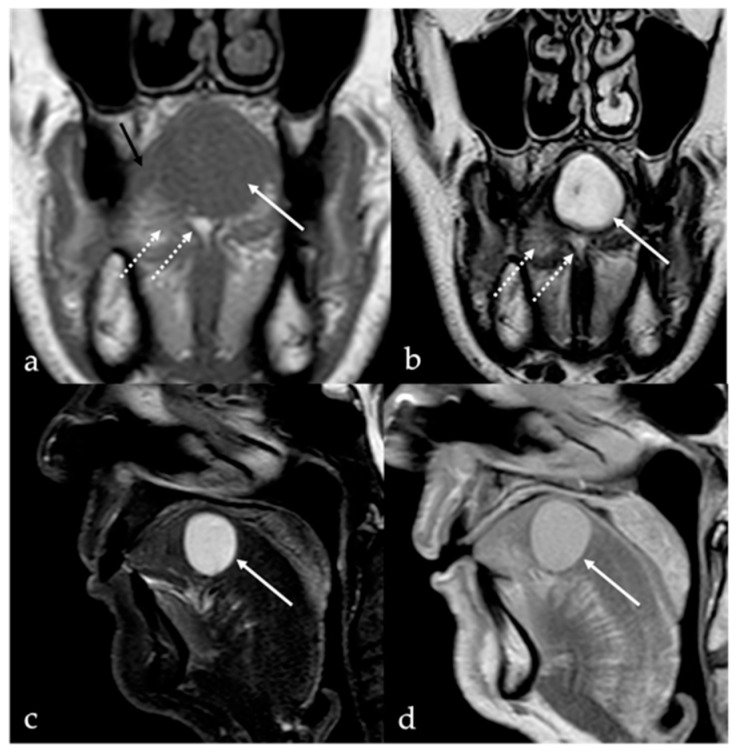
Schwannoma of the tongue in a 60-year-old male patient. Magnetic resonance imaging (MRI) reveals a submucosal rounded mass (white arrows) with well-defined margins on the midline of the mobile tongue, displacing the surrounding intrinsic tongue muscles (black arrow in (**a**)) and the adipose tissue of both lingual body and septum (white dotted arrows in (**a**,**b**)). The lesion has intermediate signal intensity (SI) on T1 (**a**), being isointense to the adjacent muscle tissue (black arrow), and homogeneous high SI on T2 (**b**) images. Because of the absence of intralesional fat tissue, Short Tau Inversion Recovery (STIR) sequence demonstrates high SI of the mass with no signal saturation (**c**). After injection of the gadolinium-based contrast agent, the mass shows rapid, vivid, and homogeneous contrast enhancement (**d**).

**Figure 2 cancers-17-01695-f002:**
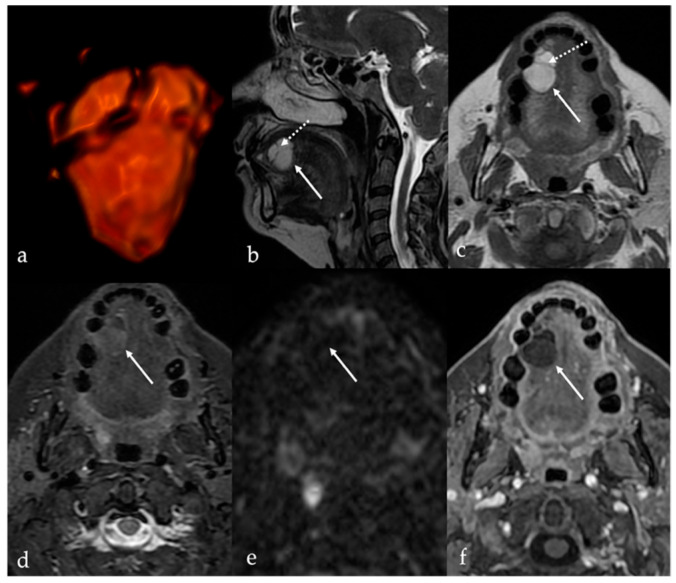
Simple lipoma of the lingual body in a 59-year-old female patient. Magnetic resonance imaging (MRI) reveals an oval, well-defined mass (white arrows) located in the right side of the mobile tongue close to the right margin and the lingual apex. The mass (represented on direct volume rendering in (**a**)) shows intralesional fine septa (white dotted arrows) and high signal intensity (SI) on T2 sequence (**b**,**c**), homogenous intralesional signal saturation on Short Tau Inversion Recovery (STIR) sequence (**d**), lack of SI on diffusion-weighted imaging (**e**), and absence of enhancement after intravenous injection of gadolinium-based contrast medium (**f**), resembling subcutaneous adipose tissue.

**Figure 3 cancers-17-01695-f003:**
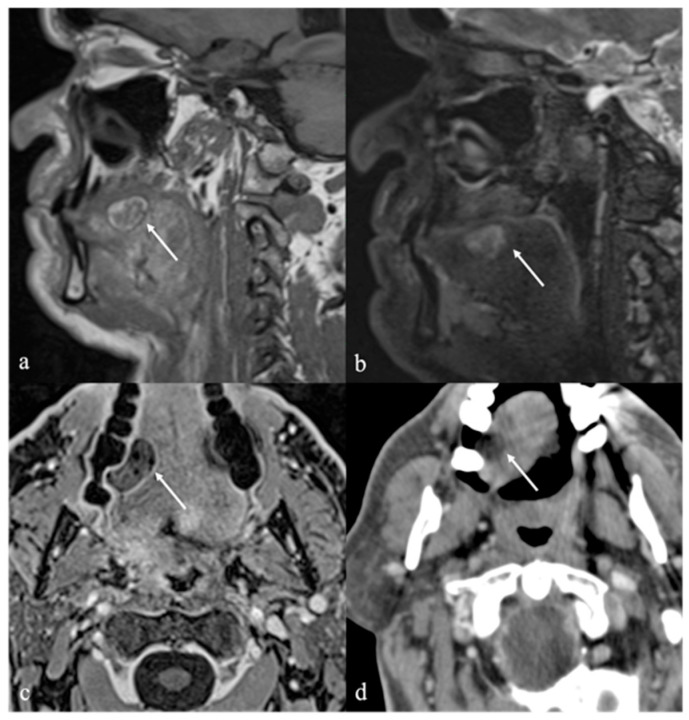
Angiomyolipoma of the lingual body in a 61-year-old male patient (white arrows). Magnetic resonance imaging shows an oval-shaped lesion in the right side of the posterior mobile tongue. Due to the presence of intralesional muscular–vascular components mixed with adipose tissue, the lesion is characterized by heterogeneous signal intensity on T1 (**a**) and T2 images (**b**), with faint and inhomogeneous enhancement after gadolinium contrast media injection (**c**). Intralesional adipose tissue is well depicted on computer tomography imaging as low-density areas with Hounsfield units (HU) like subcutaneous fat (−83 to −134 HU) (**d**).

**Figure 4 cancers-17-01695-f004:**
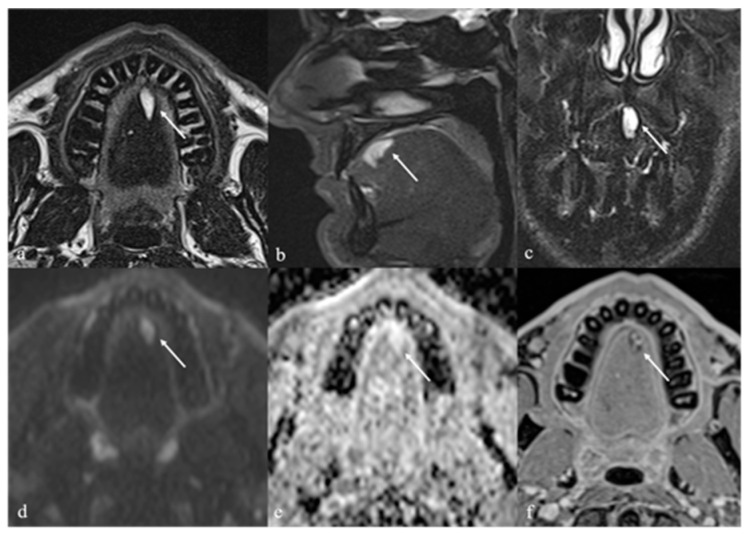
Venous malformation of the mobile tongue in a 36-year-old male patient (white arrows). Magnetic resonance imaging (MRI) depicts a centimetric, oval-shaped lesion in the left tongue located close to the lingual apex and next to the midline. The mass shows well-defined margins, high signal intensity (SI) on T2 (**a**), fat-suppressed (**b**,**c**) and diffusion-weighted imaging sequences (**d**), with high values in apparent diffusion coefficient map (**e**). The venous malformation has homogeneous filling following intravenous injection of gadolinium-based contrast media with the presence of millimetric intralesional flow voids compatible with phleboliths (**f**).

**Figure 5 cancers-17-01695-f005:**
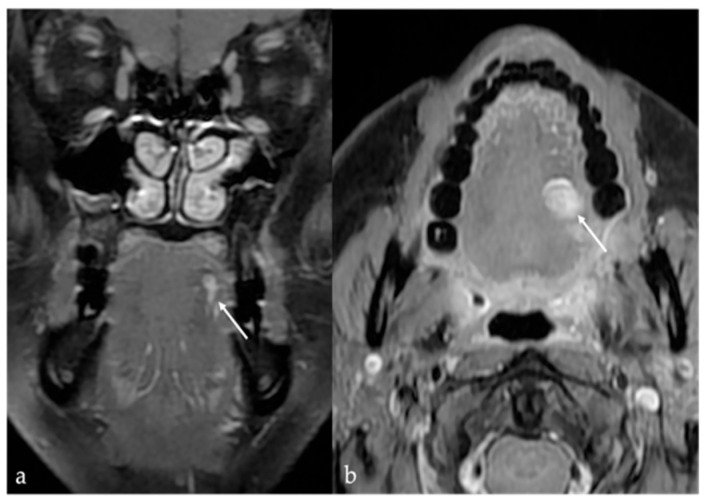
Venous malformation of the mobile tongue in a 41-year-old female patient (white arrows). Post-contrast MRI obtained with different delays (four minutes in (**a**) and ten minutes in (**b**)) from the intravenous administration of gadolinium contrast media shows a progressive and homogeneous filling of the lesion.

**Figure 6 cancers-17-01695-f006:**
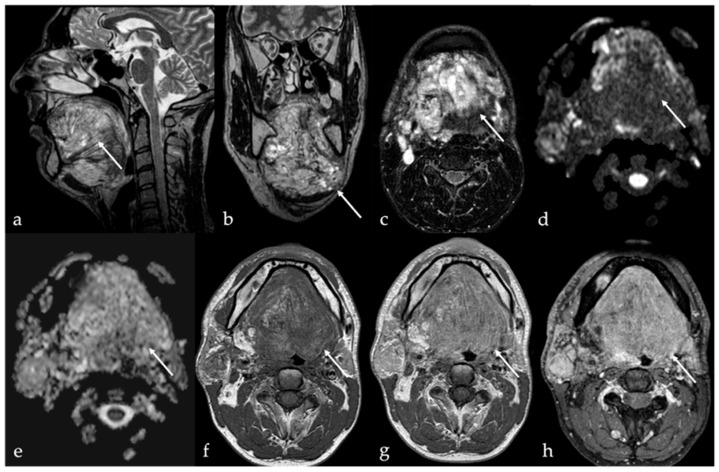
Microcystic lymphatic malformation of the tongue in a 36-year-old male patient (white arrows). Magnetic resonance imaging shows a huge, multiloculated, and microcystic mass with “trans-spatial” extension into the floor of the oral cavity, tongue, sublingual space bilaterally, right parotid space, and right carotid space. The lesion, with relatively indistinct margins, shows inhomogeneous high fluid signal intensity (SI) on T2 sequences with (**a**,**b**) and without (**c**) adipose signal saturation, and low SI on diffusion-weighted imaging (**d**) without diffusion restriction of water motion on apparent diffusion coefficient map (**e**). Pre- (**f**) and post-contrast (**g**,**h**) MRI sequences show inhomogeneous and diffuse enhancement, in absence of discernible enhancing solid components, referred to as a combined venolymphatic malformation.

**Figure 7 cancers-17-01695-f007:**
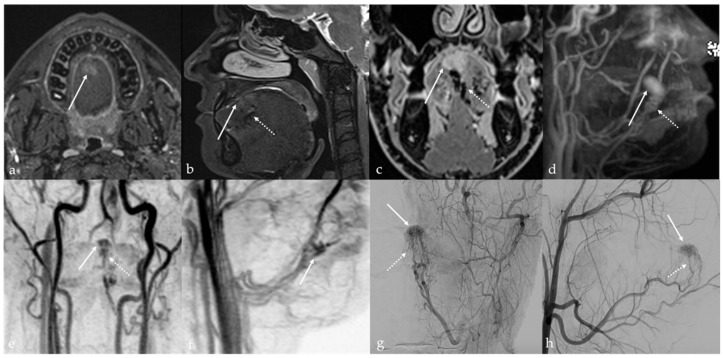
Arteriovenous malformation (AVM) in a 16-year-old male patient represented with the typical “nidus” (white arrows) and the arterial feeding vessels (white dotted arrows). Magnetic resonance imaging shows a lesion in the dorsal anterior aspect of the body tongue that is characterized by high signal intensity (SI) on T2 fat-saturated images (**a**,**b**), and huge flow voids representing anomalous arterial feeding vessels (**b**,**c**). Rapid, vivid, and homogeneous enhancement of the lesion is seen after intravenous administration of gadolinium contrast agent (**c**). Time-resolved 4D contrast-enhanced magnetic resonance angiography (4D CEMRA) maximum-intensity projection (MIP) radial reconstruction obtained in the arterial phase (**d**–**f**), and digital subtraction angiography (**g**,**h**) representing the AVM as a vascular high-flow lesion with arterial feeding vessels.

**Figure 8 cancers-17-01695-f008:**
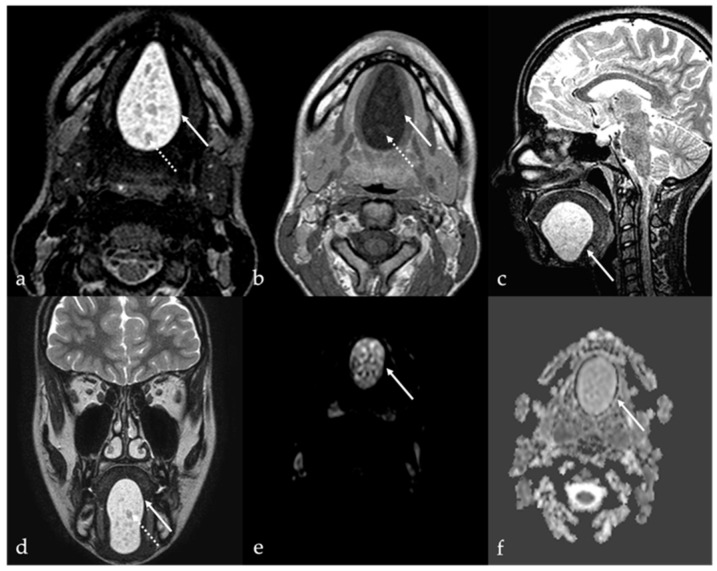
Dermoid cyst of the floor of the oral cavity in a 17-year-old female patient (white arrows). Magnetic resonance imaging shows a lesion with sharp, well-defined margins, deeply localized above the geniohyoid and mylohyoid muscles, displacing the lingual body cranially. The lesion shows high signal intensity (SI) on T2 sequence (**a**,**c**,**d**) and low SI on T1 sequence (**b**). Free fat particles (“sack of marbles sign”) with high SI on T2 (**a**,**c**,**d**) and T1 sequence (**b**) are observed within the cystic lesion (white dotted arrows). Diffusion-weighted imaging (**e**) and apparent diffusion coefficient values map (**f**) do not show restricted diffusion of water molecules motion within the cystic lesion.

**Figure 9 cancers-17-01695-f009:**
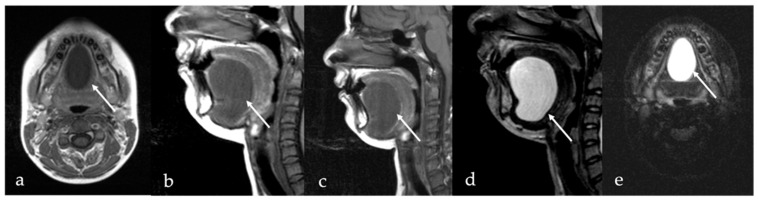
Lingual epidermoid cyst in a 16-year-old female patient. Magnetic resonance imaging T1 pre-contrast images on axial (**a**) and sagittal (**b**) planes demonstrates an oval shaped lesion with well-defined margins (white arrows) in the midline ventral tongue, deeply localized above the geniohyoid and mylohyoid muscles, manifesting with lower signal intensity (SI) than adjacent muscles. Post-contrast sagittal T1 image (**c**) shows no enhancement of the mass. The lesion has high SI on sagittal (**d**) T2 fat-suppressed sequence, with no evidence of internal fat signal. Diffusion-weighted imaging shows high SI on axial b800 image (**e**), a feature compatible with restricted diffusion of water molecules motion within the cystic lesion.

**Figure 10 cancers-17-01695-f010:**
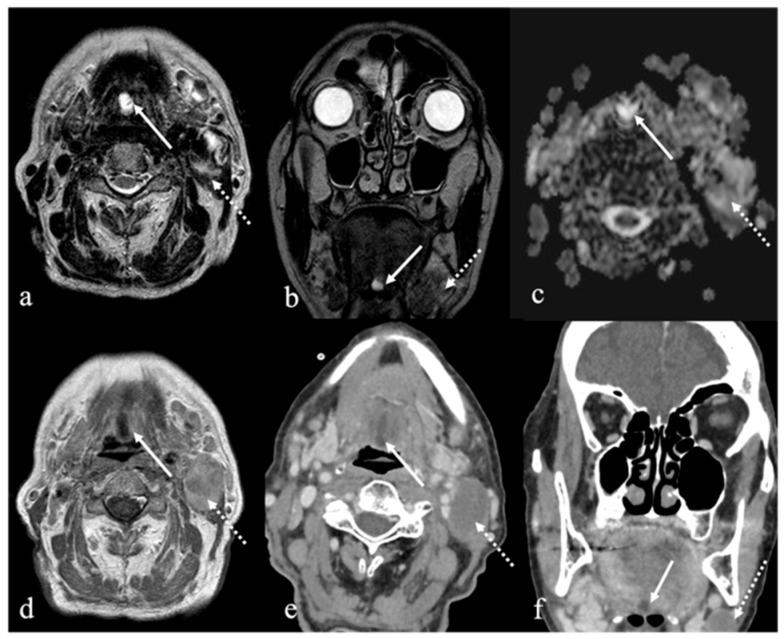
Thyroglossal duct cysts (white arrows) as an incidental finding in an 80-year-old female patient with left cervical cystic adenopathy (white striped arrows). Magnetic resonance imaging (MRI) and computed tomography (CT) show a small, round, well-defined, and median cystic lesion located at the root of the tongue, along the course of the thyroglossal duct. On MRI, the cyst shows high signal intensity (SI) on T2 (**a**,**b**), no restricted diffusion of water molecule motion on apparent diffusion coefficient values map (**c**), and low high SI on T1 (**d**). The lesion appears homogeneously hypodense on CT (**e**,**f**). No contrast enhancement is found within the lesion (**d**–**f**).

**Figure 11 cancers-17-01695-f011:**
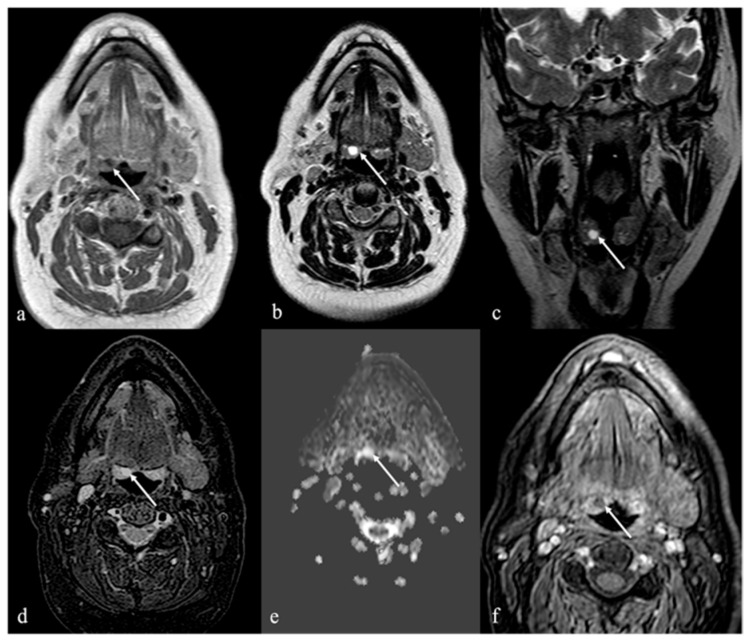
Lingual tonsil retention cyst in a 63-year-old female patient. Magnetic resonance imaging shows a round, well-defined, cystic lesion in the right lingual tonsil just adjacent to the foramen cecum (white arrows). The cyst shows low signal intensity (SI) on T1 (**a**), and high SI on T2 (**b**,**c**) and Short Tau Inversion Recovery sequences (**d**), without intralesional restricted diffusion of water molecules motions on apparent diffusion coefficient values map (**e**). No contrast enhancement is found within the lesion (**f**).

**Figure 12 cancers-17-01695-f012:**
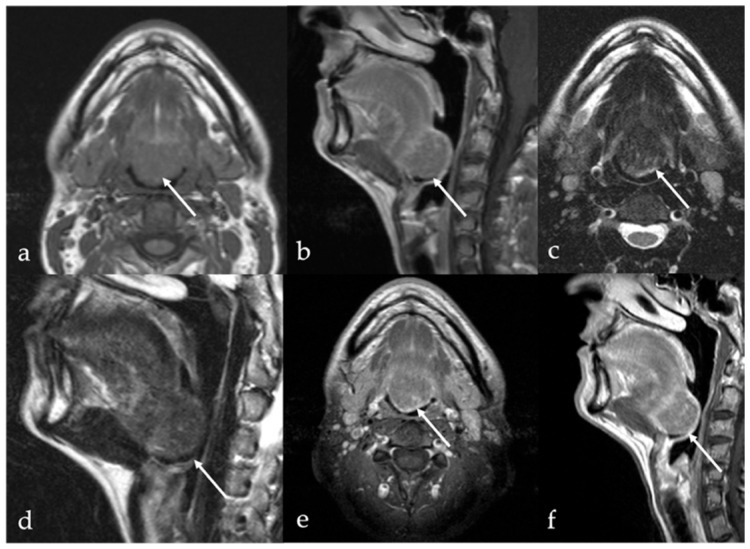
Ectopic lingual thyroid in a 35-year-old male patient. Pre-contrast T1 (**a**,**b**) and T2 (**c**,**d**) magnetic resonance imaging (MRI) sequences on axial and sagittal planes depicts a midline, well-defined soft tissue mass (white arrows) located at the tongue base that shows low signal intensity (SI) like tongue muscle. Post-contrast T1 images on axial (**e**) and sagittal (**f**) planes reveal intense, homogeneous enhancement, characteristic of functioning thyroid tissue.

**Figure 13 cancers-17-01695-f013:**
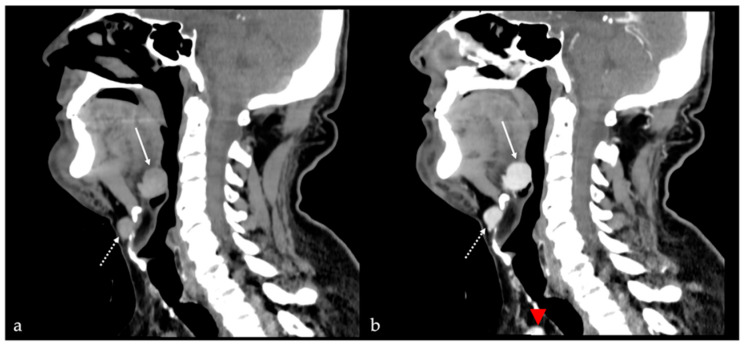
Ectopic lingual thyroid in a 58-year-old male patient. Computed tomography (CT) images show a round, well-defined, midline lesion at the base of the tongue (white arrows). The nodule appears solid and slightly hyperdense on pre-contrast CT scan (**a**). After intravenous iodine contrast media injection, the nodule shows vivid and homogeneous enhancement (**b**). Another solid nodule (white dotted arrows) of ectopic thyroid tissue is localized on the midline in the pre-epiglottic space, just antero-inferiorly to the hyoid bone. Both nodules show similar post-contrast enhancement as the thyroid gland which is in its orthotopic position (red arrowhead in (**b**) at the lower border of the field of view).

**Figure 14 cancers-17-01695-f014:**
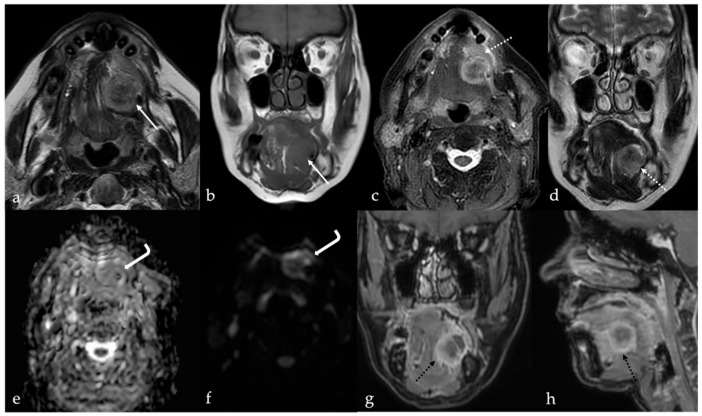
Lingual abscess in a 68-year-old woman who recently underwent surgery for squamous cell carcinoma of the left tongue margin. The abscess, located at the left lingual margin with extension into the ipsilateral sublingual space, shows inhomogeneous high signal intensity (SI) on T2 ((**a**), white arrow) and SI like the tongue muscle on T1 ((**b**), white arrow) sequences. Concomitant cellulitis edema of adjacent soft tissues in the ipsilateral sublingual space is also seen (white dotted arrows in (**c**,**d**)). The central portion (core) of the lingual abscess (white curved arrows in (**e**,**f**)) shows high restriction of the diffusion motion of water molecules with low values in the apparent diffusion coefficient map (**e**), and high signal intensity in the diffusion-weighted imaging sequence (**f**). After intravenous administration of gadolinium-based contrast media, the abscess shows an evident peripheral rim (black dotted arrows) and a relative lack of enhancement of the central core (**g,h**).

**Figure 15 cancers-17-01695-f015:**
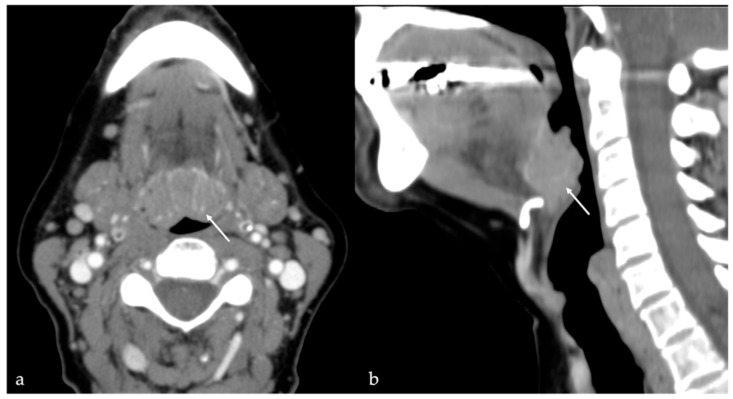
Lingual tonsillitis in a 64-year-old female patient with dysphagia and serologically proven infectious mononucleosis. Computed tomography scan acquired after intravenous administration of iodine contrast media shows a huge bilateral lingual tonsil enlargement (white arrows) that obliterates the epiglottic valleculae (**a**) and compresses the body of the epiglottis (**b**). No adenopathy is found. The axial view best depicts the symmetrical enlargement of the tongue base and the “linear pattern” of enhancement representing the lingual tonsil septa, which are typical features of benign tonsillar hypertrophy (**a**).

**Figure 16 cancers-17-01695-f016:**
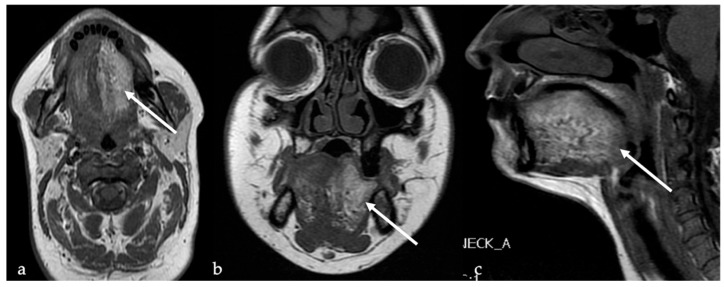
Chronic hemilingual atrophy due to unilateral hypoglossal nerve denervation (Tapia’s syndrome) in a 42-year-old female patient. Magnetic resonance imaging on axial (**a**), coronal (**b**), and sagittal (**c**) planes shows marked fatty replacement (white arrows) of the intrinsic and extrinsic muscles of the left hemilingual compartment, with increased signal intensity on non-contrast T1 images. There is a clear volume reduction in the affected left tongue side, with a mild compensatory convexity of the contralateral margin.

**Table 1 cancers-17-01695-t001:** Summary of World Health Organization (WHO) histological classification of benign tumors and tumor-like lesions of tongue, adapted from [1].

Histologic Type	Tumor/Tumor-like Lesion	Relative Frequency	Typical Tongue Location
Epithelial tumors	Papilloma	Common	Tongue base surface; mucosal
Soft tissue and neural tumors/lesions	Hemangioma Schwannoma Neurofibroma Rhabdomyoma	Common Uncommon Rare Rare	Dorsal surface; submucosal Tongue base; submucosal Unknown/none; submucosal Tongue base; submucosal
Salivary gland tumors	Pleomorphic adenoma	Rare	Unknown/none
“Dermoid” cysts	Epidermoid cyst Dermoid cyst Teratoma	Rare Rare Rare	Tongue base, submucosal Tongue base, submucosal Tongue base, submucosal
Thyroglossal duct remnants	Thyroglossal duct cyst Ectopic thyroid tissue	Common Rare	Tongue base, midline Tongue base, midline
Others ^(a)^	Lipoma Angiomyolipoma Vascular malformation Abscess Mononucleosis	Rare Rare Uncommon Very rare Very rare	Lateral tongue edge, submucosal Lateral tongue edge, submucosal Ventral tongue; submucosal Mobile tongue; submucosal Lingual tonsils

(a) Other lesions that are not included in the WHO histological classification of benign tumors and tumor-like diseases of the tongue.

**Table 2 cancers-17-01695-t002:** Main CT and MRI features of benign tumor and tumor-like lesions.

Benign Tumor/Tumor-like Lesion	Radiological Features	Differential Diagnosis
Schwannoma [6,7]	CT: hypodense MRI: high T2 SI, split fat, target and fascicular signs	Venous malformation, dermoid cysts, lipoma
Lipoma [8] Angiomyolipoma	CT: −83 to −134 HU (like subcutaneous fat tissue) MRI: high T1 and T2 SI, fat saturation signal	Chronic hemilingual denervation
Venous malformation [1,9]	CT: phleboliths MRI: high T2 SI, +CE	Other vascular malformations, schwannoma, dermoid cysts
Lymphatic malformation [10,11]	Unilocular or multilocular, microcystic (<1 cm) or macrocystic (>1 cm) CT, MRI: no solid nodule with +CE MRI: high T2 SI, fluid-fluid levels	Other vascular malformations, dermoid cysts
Arteriovenous malformation [6,12]	MRA: arterial feeding vessel, nidus, and venous drainage vessels MRI: flow voids	Other vascular malformations
Dermoid cyst [1,13,14]	CT: free fat and calcified corpuscles (“sack of marbles” sign)	Vascular malformation, epidermoid cysts, lipoma
Epidermoid cyst [14]	MRI: high SI on DWI, and restricted diffusion with low values on ADC map	Vascular malformation, dermoid cysts
Thyroglossal duct cyst [15] Ectopic thyroid tissue [16]	Cyst Same features as thyroid tissue	Lingual tonsil mucous retention cyst Squamous cell carcinoma and lymphoma of the tongue base
Abscess [17]	CT, MRI: peripheral enhancement MRI: core of the lesion shows high SI on DWI, and restricted diffusion with low values on ADC map	Submucosal malignancy
Lingual tonsillitis in infectious mononucleosis [18]	CT, MRI: symmetric tonsil enlargement, retention cysts, and “linear or columnar pattern” of enhancement (features of benign tonsillar hypertrophy)	Lingual tonsil lymphoma
Chronic hemilingual denervation [19]	Fatty atrophy of the affected hemilingual muscles with volume loss and lingual deviation, without mass-like lesion CT: −83 to −134 HU (like subcutaneous fat tissue) MRI: high T1 and T2 SI, fat saturation signal	Lipoma, angiomyolipoma

ADC: apparent diffusion coefficient; CE: contrast enhancement; CT: computed tomography; DWI: diffusion-weighted imaging; HU: Hounsfield unit; MRA: magnetic resonance angiography; MRI: magnetic resonance imaging; SI: signal intensity; T1: longitudinal relaxation time; T2: transverse relaxation time.

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
