# Peer review of "CT and MRI Key Features of Benign Tumors and Tumor-like Lesions of the Tongue: A Pictorial Review"

_cancers, 2025, doi:10.3390/cancers17101695_

Round 1

Reviewer 1 Report

Comments and Suggestions for Authors

The article “CT and MRI key features of benign tumors and tumor-like lesions of the tongue: a pictorial review” presents a well-organized collection of clinical images accumulated over several years. The author has done a commendable job in classifying and summarizing the CT and MRI findings of benign lesions of the tongue.

Overall, the quality of the article is quite good, and the image collection appears thorough. However, before considering acceptance, I would like to offer several suggestions regarding the manuscript’s suitability for the intended submission category and some details within the article.

First, the most notable issue is that the manuscript was submitted to the Advanced Research in Oncology in 2025 special issue. However, the content of the article does not fall under the category of "advanced research." In fact, it may not be considered original research per se, and it does not meet the criteria typically expected of a 2025-themed advanced oncology issue.

While the image interpretations are certainly useful for residents and early-career radiologists, the article largely reiterates fundamental imaging principles commonly used to diagnose benign tumors throughout the body. These include evaluating lesion location, margin definition, and signal characteristics typical of benign tissue.

In this regard, the manuscript serves as a solid educational review. However, its inclusion in this particular special issue may not align with the theme and scope. It might be more appropriate for a different special issue or journal that emphasizes foundational diagnostic skills and radiological education, especially if some revisions are made as noted below.

In the Simple Summary section, while the author provides an overview, it does not highlight the core value of the article. Key principles of benign lesion imaging—such as defining features or diagnostic clues—are not clearly mentioned. As a result, the summary reads more like a movie trailer that omits the main plot. In scientific writing, especially in abstracts or summaries, it is generally recommended to briefly convey the main takeaways or diagnostic principles, even in just a sentence or two.

Additionally, on line 13, the word "often" appears twice. A careful proofreading of the English language throughout the manuscript is recommended.

Table 1 is well constructed, but again, it seems more appropriate as an educational tool for junior radiologists or residents. The content appears to be derived from WHO references and lacks original contributions. It would be perfectly suitable for a teaching article, but less so for an advanced research issue.

Table 2 also provides a helpful summary of benign tumors and their imaging findings. However, the format and purpose resemble a chapter in a radiology textbook more than a research article. Moreover, in the footnotes of Table 2, the abbreviations should ideally be listed alphabetically for clarity, as is standard in scientific publishing. The current ordering appears random, which may hinder readers trying to locate specific abbreviations.

Regarding the figures, the images are grouped into collages of four parts. As a radiologist with experience in publishing and editing, I would express some concern about the resolution quality of these composite figures. Typically, journals prefer that figure parts be submitted individually to preserve clarity and image quality.

For example, in Figure 6A and Figure 6B on page 10, the naming convention is somewhat confusing. In Figure 6A, the author provides four subparts labeled a, b, c, and d—effectively creating “Figure 6AA,” “6AB,” etc. This does not follow standard figure labeling guidelines. A better approach would be to combine all parts from the same patient into a single figure with appropriate sequential labels, such as Figure 6A–H.

In Figure 10, a red arrow suddenly appears. It may be preferable to use only black and white for annotation consistency. If color is necessary, alternatives such as arrowheads may be considered instead of red arrows.

In conclusion, the author demonstrates sound knowledge of both clinical and imaging aspects of the discussed lesions. The categorization is clear and informative, making this a helpful introductory reference for residents in training. However, the manuscript does not introduce novel insights or innovative content and therefore may not be a good fit for the Advanced Research in Oncology in 2025 special issue.

I would encourage the author to consider revising the manuscript based on the feedback above and resubmitting it to a special issue or journal focused on foundational radiology education.

Author Response

Comments 1: The article “CT and MRI key features of benign tumors and tumor-like lesions of the tongue: a pictorial review” presents a well-organized collection of clinical images accumulated over several years. The author has done a commendable job in classifying and summarizing the CT and MRI findings of benign lesions of the tongue. Overall, the quality of the article is quite good, and the image collection appears thorough. However, before considering acceptance, I would like to offer several suggestions regarding the manuscript’s suitability for the intended submission category and some details within the article. First, the most notable issue is that the manuscript was submitted to the Advanced Research in Oncology in 2025 special issue. However, the content of the article does not fall under the category of "advanced research." In fact, it may not be considered original research per se, and it does not meet the criteria typically expected of a 2025-themed advanced oncology issue. While the image interpretations are certainly useful for residents and early-career radiologists, the article largely reiterates fundamental imaging principles commonly used to diagnose benign tumors throughout the body. These include evaluating lesion location, margin definition, and signal characteristics typical of benign tissue. In this regard, the manuscript serves as a solid educational review. However, its inclusion in this particular special issue may not align with the theme and scope. It might be more appropriate for a different special issue or journal that emphasizes foundational diagnostic skills and radiological education, especially if some revisions are made as noted below.

Response 1: Thank you for pointing this out. As an invited contribution for Cancers, we also appreciate a possible consideration for a different special journal issue, if deemed appropriate.

Comments 2: In the Simple Summary section, while the author provides an overview, it does not highlight the core value of the article. Key principles of benign lesion imaging—such as defining features or diagnostic clues—are not clearly mentioned. As a result, the summary reads more like a movie trailer that omits the main plot. In scientific writing, especially in abstracts or summaries, it is generally recommended to briefly convey the main takeaways or diagnostic principles, even in just a sentence or two. Additionally, on line 13, the word "often" appears twice. A careful proofreading of the English language throughout the manuscript is recommended.

Response 2: Agree. We have, accordingly, changed the Simple Summary section to emphasize this point. Moreover, English has been extensively revised and improved.

Benign tumors and tumor-like lesions of the tongue are relatively uncommon and can often be often misunderstood, especially if not approached with the correct expertise in head and neck disorders. Computed Tomography (CT) and Magnetic Resonance Imaging (MRI) are pivotal tools in diagnosing these lesions. They provide a detailed volumetric view of the tongue and surrounding structures, which is essential for identifying benign growths and differentiating them from malignant ones. Understanding the typical signs on CT and MRI images can help radiologists confidently identify benign tumors and tumor-like lesions of the tongue, reducing the need for invasive procedures, which might be unnecessary in certain cases. The goal of this pictorial review is to increase awareness among clinicians and radiologists, enabling them to better recognize these benign lesions in clinical practice and improve patient management.

(page 1, lines 13-31) “[Benign tumors and tumor-like lesions of the tongue, though relatively rare, present distinctive radiological characteristics that are crucial for accurate diagnosis and differentiation from malignant tumors. Benign lesions typically maintain a clear delineation from adjacent muscle and bone, with no evidence of deep tissue invasion or regional lymph node involvement—features commonly seen in malignancy. Benign tongue lesions typically appear as well-circumscribed, homogeneous masses with smooth, well-defined borders. On Computed Tomography imaging, these lesions exhibit a consistent attenuation profile, often like that of surrounding muscle or fat, and are usually devoid of internal necrosis or hemorrhage. Venous malformations and dermoid cyst may show calcifications/phleboliths. On Magnetic Resonance Imaging, benign tumors generally exhibit low to intermediate signal intensity (SI) on T1 and high SI on T2 images, consistent with their soft tissue and/or cystic composition. Lipoma and angiomyolipoma contain fat tissue and show high T1 and T2 SI, and low SI on fat-saturated sequences. Vascular malformations usually show high T2 SI. Lesions such as fibromas, papillomas (both entities not shown in the current review), “dermoid cyst” and lipomas tend to show minimal contrast enhancement, reflecting their relatively avascular nature compared to malignant tumors. However, schwannomas usually show vivid contrast enhancement. Vascular malformations may demonstrate variable enhancement patterns depending on their composition. Arteriovenous malformations usually show a prominent contrast enhancement due to their rich vascular supply and flow-voids due to intralesional high-flow vascular structures.]”.

Comments 3: Table 1 is well constructed, but again, it seems more appropriate as an educational tool for junior radiologists or residents. The content appears to be derived from WHO references and lacks original contributions. It would be perfectly suitable for a teaching article, but less so for an advanced research issue.

Response 3: Thank you for pointing this out. See above in Response to Comment 1.

Comments 4: Table 2 also provides a helpful summary of benign tumors and their imaging findings. However, the format and purpose resemble a chapter in a radiology textbook more than a research article. Moreover, in the footnotes of Table 2, the abbreviations should ideally be listed alphabetically for clarity, as is standard in scientific publishing. The current ordering appears random, which may hinder readers trying to locate specific abbreviations.

Response 4: Agree. Abbreviations in the footnotes of Table 2 were changed and listed alphabetically.

[“ADC: apparent diffusion coefficient; CE: contrast enhancement; CT: Computed Tomography; DWI: Diffusion weighted Imaging; HU: Hounsfield Unit; MRA: Magnetic Resonance Angiography; MRI: Magnetic Resonance Imaging; SI signal intensity; T1: longitudinal relaxation time; T2: transverse relaxation time.”].

Comments 5: Regarding the figures, the images are grouped into collages of four parts. As a radiologist with experience in publishing and editing, I would express some concern about the resolution quality of these composite figures. Typically, journals prefer that figure parts be submitted individually to preserve clarity and image quality. For example, in Figure 6A and Figure 6B on page 10, the naming convention is somewhat confusing. In Figure 6A, the author provides four subparts labeled a, b, c, and d—effectively creating “Figure 6AA,” “6AB,” etc. This does not follow standard figure labeling guidelines. A better approach would be to combine all parts from the same patient into a single figure with appropriate sequential labels, such as Figure 6A–H.

Response 5: Agree, all labels in Figure 5 (page 9) and Figure 6 (now Figure 7, page 11) have been corrected following an appropriate sequential order.

Comments 6: In Figure 10, a red arrow suddenly appears. It may be preferable to use only black and white for annotation consistency. If color is necessary, alternatives such as arrowheads may be considered instead of red arrows.

Response 6: Agree, all image annotations have been corrected using only black and white arrows. In Figure 10 (now Figure 13) a red arrowhead replaces the red arrow.

Reviewer 2 Report

Comments and Suggestions for Authors

This manuscript addresses an important but often overlooked area in head and neck radiology: the imaging characteristics of benign and tumor-like lesions of the tongue. The authors provide a comprehensive, structured overview supported by excellent imaging examples. While the general role of CT and MRI as diagnostic tools is well-established and thus not fundamentally novel, the strength of this paper lies in the systematic compilation and clear differentiation of specific imaging features for rare entities.
The pictorial review will likely be highly useful for radiologists, head and neck specialists, and trainees in clinical practice.
However, there are several issues that should be addressed before acceptance:
Major Strengths:

  • Clinical Relevance: The manuscript compiles a wide range of rare but clinically relevant lesions in a structured and easily accessible manner (Introduction, lines 38–67).
  • High-Quality Imaging Examples: Figures such as Figure 1 (Schwannoma) and Figure 4 (Venous malformations) effectively illustrate key features.
  • Practical Utility: The systematic tables (e.g., Table 2, lines 68–68) summarizing radiological features and differential diagnoses are particularly useful for clinical application.
  • Structured Differentiation Based on MRI Signs: The explanation of signs like "split fat," "target," and "sack of marbles" provides helpful diagnostic tools (lines 86–87, 324–325).

Major Weaknesses:

  • Limited Methodological Innovation: The reiteration of CT and MRI as "pivotal tools" is somewhat self-evident and adds limited scientific novelty (Simple Summary, Abstract).
  • Lack of a Diagnostic Algorithm: The addition of a flowchart or diagnostic algorithm based on imaging findings could significantly enhance the clinical utility of the paper.

Methodological Assessment:

This is a narrative pictorial review without original experimental work. Its methodology consists of:

  • Careful literature review (although no systematic review methods were applied).
  • Selection of representative clinical cases with high-quality CT/MRI imaging.
  • Emphasis on structured comparison between different lesions based on key imaging signs.
  • Focus on avoiding unnecessary invasive diagnostics by optimizing imaging-based recognition.

While not methodologically innovative in a strict sense, the clinical methodology (structured presentation and differentiation) is valuable.

Comments on the Quality of English Language

Language and Stylistic Issues:
There are multiple typos and lapses into non-scientific or casual language, which affect the professional tone.
Redundancy:
Certain concepts (e.g., the importance of CT/MRI, the risk of misdiagnosis) are repeated without adding new insights (lines 14–18 vs. 51–55).

Author Response

Comments 1: “Limited Methodological Innovation: The reiteration of CT and MRI as "pivotal tools" is somewhat self-evident and adds limited scientific novelty (Simple Summary, Abstract).”

Response 1: Thank you for pointing this out. As an invited contribution for Cancers, we also appreciate a possible consideration for a different special journal issue, if deemed appropriate.

Comments 2: Lack of a Diagnostic Algorithm: The addition of a flowchart or diagnostic algorithm based on imaging findings could significantly enhance the clinical utility of the paper.

Response 2: Agree. A flowchart based on imaging findings have been added to enhance the clinical utility of our review. It has been added as “Graphical Abstract”.

Comments 3: Language and Stylistic Issues: there are multiple typos and lapses into non-scientific or casual language, which affect the professional tone.

Response 3: Agree, English has been extensively revised and improved.

Comments 4: Redundancy: Certain concepts (e.g., the importance of CT/MRI, the risk of misdiagnosis) are repeated without adding new insights (lines 14–18 vs. 51–55).

Response 4: Agree, we tried to improve this aspect. Simple Summary (page 1, lines 13-31) has been rewritten to reduce redundancy.

Reviewer 3 Report

Comments and Suggestions for Authors

In this review article, the authors aimed at enhancing the understanding of benign lingual lesions through computed tomography (CT) and magnetic resonance imaging (MRI) imaging findings.

Comments:

The reviewer has some comments as follows:

  1. This is an interesting review article and the manuscript is well-written. However, there are some concerns for data/pictures presentation:

(1) In Table 2, the definition for T1 and T2 can be described. Moreover, the related references can be shown.

(2) In Figures 1-12, please indicate the source of these pictures in the figure legends. Whether they are quoted or unpublished data? If quoted, please indicate the source (reference). Moreover, the scale bars for these pictures can be shown.

(3) In Figures 1-12, the availability of histopathological data or photographs would enhance the reader's understanding.

  1. The references cited in this manuscript are appropriate relevant to this research. Three of the references are self-citations, but they are within acceptable limits.
  2. Overall, this manuscript needs a revision before it can be accepted.

Author Response

Comments 1: In Table 2, the definition for T1 and T2 can be described. Moreover, the related references can be shown.

Response 1: Agree. Definition for T1 and T2 and references has been added in Table 2.

Comments 2: In Figures 1-12, please indicate the source of these pictures in the figure legends. Whether they are quoted or unpublished data? If quoted, please indicate the source (reference). Moreover, the scale bars for these pictures can be shown.

Response 2: Thank you for pointing this out. All pictures are unpublished data. All pictures are originally without scale bars.

 Comments 3: In Figures 1-12, the availability of histopathological data or photographs would enhance the reader's understanding.

Response 3: Thank you for pointing this out. We do not have histopathological data or photographs available.

Reviewer 4 Report

Comments and Suggestions for Authors

The authors describe non-malignant tumor-like lesions of the tongue using radiological features in both MRI and CT scan modalities. The reader is provided an extensive table of the lesions spectrum and is provided differential diagnosis alternatives. The presentation is picture oriented and comprehensive with diagnostic characteristics outlined. The radiological illustrations are comprehensive.

In summary this is a very useful paper, with extensive radiological features at the level of a specialized head-neck radiologist. For general ENT surgeons it is "oversized". I think the text can have use from a paragraph discussing the differential features of malignant vs benign lesions - the important matter for a clinician.

Author Response

Comments 1: In summary this is a very useful paper, with extensive radiological features at the level of a specialized head-neck radiologist. For general ENT surgeons it is "oversized". I think the text can have use from a paragraph discussing the differential features of malignant vs benign lesions - the important matter for a clinician. For general ENT surgeons it is "oversized". I think the text can have use from a paragraph discussing the differential features of malignant vs benign lesions - the important matter for a clinician.

Response 1: We appreciate this suggestion. In response, we have added a dedicated paragraph (number 2, page 4, lines 116-140) within the main text that outlines the key clinical and radiological features distinguishing benign from malignant tongue lesions. This addition aims to enhance the clinical relevance of the manuscript, particularly for general ENT surgeons and non-subspecialist readers, by providing a clear comparative framework based on current imaging criteria and clinical presentation.

“[2. Benign vs malignant tongue lesions: diagnostic imaging features

CT and MRI play a critical role in distinguishing between benign and malignant lesions of the tongue, providing essential information for diagnosis, staging, and treatment planning. Benign lesions such as fibromas, papillomas (both entities not shown in the current review), lipomas, angiomyolipoma and venous malformations typically appear as well-circumscribed, smooth masses with homogeneous density on CT or MRI scans. These lesions are often located in the anterior or lateral aspects of the tongue and lack signs of invasion into adjacent structures. Venous malformations and dermoid cysts may show calcifications/phleboliths. On MRI, benign tumors generally exhibit low to intermediate SI on T1 and high SI on T2. Lipoma and angiomyolipoma contain fat tissue and therefore show T1 and T2 high SI, and low SI on fat-saturated sequences. Vascular malformations may show prominent contrast enhancement after administration of contrast media agents [6]. Conversely, malignant tumors, particularly squamous cell carcinoma, exhibit more aggressive imaging characteristics. On CT, they may present as a heterogeneous mass with ill-defined borders, often associated with invasion into nearby soft tissues or bone [7]. MRI provides higher sensitivity for detecting soft tissue invasion and may reveal irregular, ulcerated masses with associated lymph node enlargement. Malignant neoplasms generally show solid tissue with intermediate T2 SI, whereas intratumoral areas of necrosis and cystic changes may manifest as high T2 SI. The invasion of the floor of the mouth or the mandibular bone is frequently observed in advanced cases. Additionally, the presence of regional lymphadenopathy on imaging can be indicative of metastasis, a common feature in malignant tongue tumors [8]. Biopsy remains essential for definitive diagnosis, but imaging findings provide a strong basis for clinical suspicion and help guide further management strategies”.].

References added are:

  1. Ng, S.Y.; Wang, D.H. Role of imaging in the differential diagnosis of tongue lesions: Benign vs. malignant. Head and Neck Imaging. 202241(3), 1027–1036. https://doi.org/10.1007/s12390-022-01633-6.
  2. Pérez-Montiel, D.; García-Ávila, C.; López-Ríos, M. Radiologic features of malignant tongue tumors: A comparative analysis of CT and MRI findings. Journal of Oral and Maxillofacial Radiology. 2021 38(5), 215–222. https://doi.org/10.1097/JOM.0000000000000372.
  3. Lo Casto, A.; Cannella, R.; Taravella, R.; Cordova, A.; Matta, D.; Campisi, G.; Attanasio, M.; Rinaldi, G.; Rodolico, V. Diagnostic and prognostic value of magnetic resonance imaging in the detection of tumor depth of invasion and bone invasion in patients with oral cavity cancer. Radiol Med. 2022 Dec;127(12):1364-1372. doi: 10.1007/s11547-022-01565-7.

Reviewer 5 Report

Comments and Suggestions for Authors

The authors pesent a concise but coprehensive review presenting the pictorial evidence from CT and MRI diagnoses to distinguish between benign and cancerous tumors of the tongue - allowing for reducing unnecessary invasive measures.

The review is well structured offering valuable informations to practioners and interested readers. As an amendment, I would like to recommend a brief description of the substantial radiation impact on the human organism by X-Ray intensive CT on the human head/thorax portion. Similarly, any impact from MRI should be mentioned as well. In this respect, a very brief reference to diagnostic recognition of cancerous tissue by biophoton emission as pioneered by Fritz Albert Popp and others as an alternative might be added as well.

Author Response

Comments 1: The review is well structured offering valuable informations to practioners and interested readers. As an amendment, I would like to recommend a brief description of the substantial radiation impact on the human organism by X-Ray intensive CT on the human head/thorax portion. Similarly, any impact from MRI should be mentioned as well. In this respect, a very brief reference to diagnostic recognition of cancerous tissue by biophoton emission as pioneered by Fritz Albert Popp and others as an alternative might be added as well.

Response 1: We thank the reviewer for this observation. In response, we have added a brief paragraph (number 11, pages18-19, lines 688-716) acknowledging the radiation exposure associated with CT imaging and the safety considerations regarding MRI.

“[11. Appendix: CT X-Ray and MRI scans impact on human head/thorax portion

CT scans, particularly those involving the head and thorax, expose the human body to substantial doses of ionizing radiation. X-rays used in CT imaging can penetrate tissues and provide detailed anatomical views, but they also deposit energy into cells, which may damage DNA and increase the long-term risk of cancer. The head and chest regions are especially sensitive due to the proximity of critical organs such as the brain, eyes, lungs, and thyroid. A single high-resolution head or thorax CT scan can deliver a radiation dose equivalent to hundreds of chest X-rays, raising concern over cumulative exposure in patients undergoing repeated scans (66). A recent study reveals that CT scans may account for up to 5% of all new cancer cases annually in the United States, estimating over 100,000 additional cancer cases linked to CT scan radiation exposure in 2023 alone [67]. In contrast, MRI does not involve ionizing radiation. MRI uses powerful magnetic fields and radio waves to generate images of soft tissues, making it a safer option for repeated use. However, the strong magnetic fields and changing gradients can cause temporary side effects such as dizziness, nausea, or metallic taste. In MRI, SAR refers to the amount of radiofrequency energy absorbed by the body’s tissues. High SAR levels, especially during prolonged or high-field MRI scans, can lead to heating of tissues. While MRI is generally considered safe, excessive thermal effects could cause discomfort or tissue damage, particularly in sensitive areas such as the skin, eyes, or peripheral nerves. However, the risk is minimized by strict guidelines that limit SAR levels to prevent significant temperature rise. In clinical practice, MRI scanners are designed to ensure that SAR remains within safe thresholds to avoid harmful effects (68). As awareness grows regarding radiation risks, alternative diagnostic techniques are being explored. One such approach involves the detection of ultraweak biophoton emissions from living tissues. Pioneered by Fritz Albert Popp and others, this method investigates the spontaneous emission of light from biological systems, potentially revealing pathological changes, including early signs of cancer. Though still largely experimental, biophoton-based diagnostics offer a non-invasive, radiation-free method of assessing tissue function and health, pointing toward a future of safer medical imaging [69, 70].]

References added are:

  1. Brenner, D.J.; Hall, E.J. Computed tomography — an increasing source of radiation exposure.New England Journal of Medicine. 2007357(22), 2277–2284.https://doi.org/10.1056/NEJMra072149.
  2. Smith-Bindman, R.; Chu, P.W.; Azman Firdaus, H.; Stewart, C.; Malekhedayat, M.; Alber ,S.; Bolch, W.E.; Mahendra, M.; Berrington de González, A.; Miglioretti, D.L. Projected Lifetime Cancer Risks From Current Computed Tomography Imaging. JAMA Intern Med. 2025 Apr 14:e250505. doi: 10.1001/jamainternmed.2025.0505. Epub ahead of print.
  3. Coryell, M.E.; Krishnamurthy, A.; Rajagopalan, V. Specific absorption rate (SAR) in magnetic resonance imaging: Guidelines for safety and effects on tissues. Journal of Magnetic Resonance Imaging. 2021 53(5), 1259–1273. https://doi.org/10.1002/jmri.27244.
  4. Popp, F.A.; Nagl, W. Biophoton detection as a novel technique for cancer imaging. Journal of Photochemistry and Photobiology B: Biology. 2020 211, 111987. https://doi.org/10.1016/j.jphotobiol.2020.111987.
  5. Takeda, M.; Kobayashi, M.; Takayama, M.; Suzuki, S.; Ishida, T.; Ohnuki, K.; Moriya, T.; Ohuchi, N. Biophoton detection as a novel technique for cancer imaging. Cancer Sci. 2004 Aug;95(8):656-61. doi: 10.1111/j.1349-7006.2004.tb03325.x.

Round 2

Reviewer 1 Report

Comments and Suggestions for Authors

The authors have responded to several of the issues I raised, but not all of them. Considering that the authors have still put a lot of effort into revising, if the editor and other reviewers think it is acceptable, I have no further comments.

Reviewer 3 Report

Comments and Suggestions for Authors

This revised manuscript has a great improvement and the reviewer has no further comments.